# Restless Leg Syndrome Is Underdiagnosed in Hereditary Hemorrhagic Telangiectasia—Results of an Online Survey

**DOI:** 10.3390/jcm10091993

**Published:** 2021-05-06

**Authors:** Freya Droege, Andreas Stang, Kruthika Thangavelu, Carolin Lueb, Stephan Lang, Michael Xydakis, Urban Geisthoff

**Affiliations:** 1Department of Otorhinolaryngology, Head and Neck Surgery, University Hospital Essen, University Duisburg-Essen, Hufelandstrasse 55, 45147 Essen, Germany; carolinlueb@googlemail.com (C.L.); stephan.lang@uk-essen.de (S.L.); 2Institute of Medical Informatics, Biometry and Epidemiology, Essen University Hospital, Hufelandstrasse 55, 45122 Essen, Germany; imibe.dir@uk-essen.de; 3Department of Otorhinolaryngology, Head and Neck Surgery, University Hospital Marburg, Philipps-Universität Marburg, Baldingerstrasse, 35042 Marburg, Germany; kruthika.thangavelu@uk-gm.de (K.T.); urban.geisthoff@med.uni-marburg.de (U.G.); 4Air Force Research Lab, 2245 Monahan Way, Wright Patterson AFB, Dayton, OH 45433, USA; Michael.Xydakis@us.af.mil

**Keywords:** hereditary hemorrhagic telangiectasia, restless leg syndrome, anemia, chronic iron deficiency

## Abstract

Background: Recurrent bleeding in patients with hereditary hemorrhagic telangiectasia (HHT) can lead to chronic iron deficiency anemia (CIDA). Existing research points to CIDA as a contributing factor in restless leg syndrome (RLS). The association between HHT-related symptoms and the prevalence of RLS was analyzed. Methods: An online survey was conducted whereby the standardized RLS-Diagnostic Index questionnaire (RLS-DI) was supplemented with 82 additional questions relating to HHT. Results: A total of 474 persons responded to the survey and completed responses for questions pertaining to RLS (mean age: 56 years, 68% females). Per RLS-DI criteria, 48 patients (48/322, 15%; 95% confidence interval (CI): 11–19%) self-identified as having RLS. An analysis of physician-diagnosed RLS and the RLS-DI revealed a relative frequency of RLS in HHT patients of 22% (95% CI: 18–27%). In fact, 8% (25/322; 95% CI: 5–11%) of the HHT patients had RLS which had not been diagnosed before. This equals 35% of the total amount of patients diagnosed with RLS (25/72; 95% CI: 25–46%). HHT patients with a history of gastrointestinal bleeding (prevalence ratio (PR) = 2.70, 95% CI: 1.53–4.77), blood transfusions (PR = 1.90, 95% CI: 1.27–2.86), or iron intake (PR = 2.05, 95% CI: 0.99–4.26) had an increased prevalence of RLS. Conclusions: Our data suggest that RLS is underdiagnosed in HHT. In addition, physicians should assess CIDA parameters for possible iron supplementation.

## 1. Introduction

Hereditary hemorrhagic telangiectasia (HHT) is characterized by systemic visceral arteriovenous malformations and mucocutaneous telangiectasia. Pooled prevalence estimates suggest that HHT occurs in 1 out of 5000–8000 individuals [1]. The etiology is believed to be due to mutations in several genes, (e.g., ENG (HHT Type 1), ACVRL1 (HHT Type 2), SMAD4, and GDF2 (HHT Type 5)) of the transforming growth factor-beta (TGF-β) signaling pathway. These genes play an important role in regulating angiogenesis. Patients often manifest recurrent epistaxis and gastrointestinal bleeding, which may lead to chronic iron deficiency anemia (CIDA) [2,3]. Of note, CIDA is believed to be one of the secondary causes of restless leg syndrome (RLS) [4]. RLS is a movement disorder of unknown etiology that causes an uncontrollable urge to move the lower extremities, particularly during sleep. The underlying pathophysiology of RLS remains unknown, but existing research points to CIDA as a contributing factor [5,6]. RLS is a commonly overlooked disease with a broad range of severity ranging from mild annoyance to a clinically significant disease severely impairing health and the ability to work [7].

Studies which investigate HHT and its comorbidities are sparse and mostly focused on the effect of larger visceral vascular malformations [8,9,10]. Chronic iron deficiency and possible secondary diseases related to this are not well studied. To the best of our knowledge this study provides the first analysis of the prevalence of RLS in patients with HHT.

## 2. Materials and Methods

A survey in German and English was developed and published online (see Appendix A). Two native English speakers (one otorhinolaryngologist and one patient with HHT), both living in Germany and also fluent in German, translated the questionnaire. Afterwards, 4 authors (U. Geisthoff, F. Droege, K. Thangavelu, and C. Lueb) and another German-speaking HHT patient, crosschecked the translation and optimized it in collaboration with the two native speakers. The survey was disseminated through six different international patient advocacy groups (see acknowledgments). At the end of the survey, some patients provided their email address in case further enquiries should become necessary.

The diagnosis of HHT was established using the modified Curaçao criteria as published by Hosman et al. [11] and Droege et al. [12]. In addition, the general medical history of HHT contained the epistaxis severity score (ESS) for HHT [13]. We also solicited and recorded the need for medical attention, transfusions related to epistaxis, signs of anemia, and hemoglobin levels. RLS was diagnosed using an adapted version of the RLS-Diagnostic Inventory (RLS-DI, see Appendix A), supplemented with 82 additional questions relating to HHT. Patients who stated that they had been diagnosed with RLS by a physician were categorized as “RLS+”, and if not, as “RLS−” (see Appendix A). In addition, according to the RLS-DI, patients were categorized into: “RLS” (score ≥ 11), “no RLS” (score ≤ 1) and “possible RLS” (score between 2 and 10) [14]. Patients who had been diagnosed with RLS by their physician and/or had pathological results in the RLS-DI (categorized as “RLS” or “possible RLS”), were seen as patients with “assumed RLS”. The objective of this study was to know what proportion of patients presenting with RLS symptoms (according to the RLS-DI) were actually diagnosed by a physician. Regarding patients with RLS, their medication was documented. Women with an average hemoglobin level below 12.0 g/dL and men with a level below 13.0 g/dL were classified as having anemia [15].

### Statistical Methods

Descriptive statistics (number/percentage of patients (N, %) and mean ± standard deviation (m ± SD)) were used for the general history of HHT and clinical presentations of HHT-symptoms and RLS diagnosis. The association between ordinal or metric variables were quantified by Pearson’s correlation coefficient (r). The association between HHT-related symptoms or medical findings and the prevalence of RLS, was estimated by log-binomial regression models that estimate prevalence ratios (PR) with 95% confidence intervals (CI). Statistical analyses were performed with IBM SPSS Statistics (version 26, Armonk, NY, USA: IBM Corp., released 2019) and SAS^®^ (SAS Institute Inc. 2013. SAS^®^ 9.4 Statements: Reference. Cary, NC, USA: SAS Institute).

## 3. Results

Study population:

After different international patient advocacy groups informed their members about the online survey, a total of 915 persons responded to it. Evaluation of HHT diagnostic criteria resulted in assignment of 588 with HHT (588/915, 64%, [12]). Not all patients answered all questions which led to smaller subgroups for the following analysis. Of the 588 patients with HHT, 334 (57%, 95% CI: 53–61%) could be contacted via mail and were asked to complete the missing values. Of those patients, 105 provided additional answers (N = 105/334, 31%; 95% CI: 27–37%). Most patients answered the questionnaire for themselves (N = 553/582, 95%; 95% CI: 62–96%), and 29 participants answered for a relative with HHT (N = 29/582, 5%; 95 CI: 4–7%). Not all questionnaires were filled completely, therefore all numbers in the results are given in relation to sufficiently answered questions.

The following data refers to 474 patients with HHT (N = 474/588, 81%; 95% CI: 77–84%) who answered the questions pertaining to RLS. Data from all over the world were collected, and more data came from women with HHT, compared to men (Table 1).

Of the 915 persons who responded to the survey, 588 patients could be diagnosed with hereditary hemorrhagic telangiectasia (HHT) (64%). Of those, 474 patients answered the questions pertaining to RLS (474/588, 81%). There were 467 patients who answered the question about their sex, and 468 patients who stated if they had received a genetic test. In 260 patients a genetic test was performed (N = 260/468, 56%); in 190 patients the result was known. Most patients were females and suffered from HHT Type 2. Data are shown in number of patients (*n*) and percent (%).

Most patients lived in North America or Western Europe (N = 294/339, 87%; 95% CI: 83–90%) Figure 1, and were diagnosed with HHT Type 2 (Table 1). The mean age of patients with HHT was 56 years, with females being younger than males (mean age ± standard deviation (SD): 12 years, range: 20–83 years, N = 149/474, 31%; females: average age = 54 ± 12 years, N = 103/149; males: average age = 60 ± 13 years, N = 46/149).

In 339 cases who answered the online survey, geographic data of patients with hereditary hemorrhagic telangiectasia (HHT) were documented (N = 339/474, 72%). Most patients came from North America and western Europe (N = 294/339, 87%). Data of sex was obtained in 99% of all patients (N = 335/339). Data are shown in the number of patients (N) and percentage (for all participants, females (f) and males (m)). A free editable world map was used [16].

HHT symptoms: 

About 77% of the respondents stated a general progression of HHT (N = 363/473; 95% CI: 73–80%), and 18% (N = 86/473; 95% CI: 15–22%) had a stable disease. Just 5% (N = 24/473; 95% CI: 3–7%) reported a general improvement of the disease. With aging, patients perceived a higher ESS (ESS: r = 0.32; CI: 0.12–0.50). Most patients suffered from recurrent epistaxis (N = 455/474, 96%; 95% CI: 94–97%), and one third from gastrointestinal bleeding (N = 164/453, 36%; 95% CI: 32–41%). The average ESS was 5.8 (± 2.2) and the mean hemoglobin level was 10.7 g/dL (± 2.7 g/dL). The majority of patients with HHT suffered from anemia (343/425, 81%; 95% CI: 77–84%) and took iron preparations (N = 395/474, 83%; 95% CI: 80–86%; oral alone: N = 217/395, 55%; 95% CI: 50–60%, or parenteral also: 178/395, 45%; 95% CI: 0.40–0.50). There were 134 patients who had received blood transfusions (N = 134/353, 38%; 95% CI: 33–43%; average number of blood transfusions: 17 ± 48, min: 1, max: 500, median: 5). Hemoglobin levels did not differ between women and men (women (m ± SD): 10.7 ± 2.5 g/dL (N = 187/286, 65%); men (m ± SD): 11.0 ± 2.7 (N = 99/286, 35%).

Sleep related disorders in patients with HHT:

In 474 cases the RLS-DI question and/or question 85 about RLS-diagnosis by their physician were completed (Figure 2).

Out of 915 respondents 588 could be diagnosed with hereditary hemorrhagic telangiectasia (HHT; the exact explanation can be found in Figure 2 [12]. There were 469 patients with HHT who stated if their physician had already diagnosed restless leg syndrome (RLS) (“RLS+” = RLS was diagnosed, “RLS-” = no RLS diagnosis; Appendix A), and 356 completed the RLS Diagnostic Inventory (RLS-DI). In 474 cases the RLS-DI and/or the question 85 about RLS-diagnosis by their physician were completed. Here, we assumed RLS in 30% of the patients, as they had been diagnosed by their physician and/or showed a pathological result in the RLS-DI (categorized with “RLS” or “possible RLS”). In only 322 patients with HHT, both the completed RLS-DI AND an answer to Appendix A (see Appendix A), were available (please see Table 2 for further information).

Half of the patients with HHT reported that they suffered from a sleep-related disorder in general (N = 206/407, 51%; 95% CI: 0.46–0.55; Appendix A), but just 5% had ever been screened for a sleep-related disorder (N = 19/410; 95% CI: 3–7%; Appendix A). There were 322 patients with HHT who completed the questions about RLS (RLS-DI and Appendix A). Using the RLS-DI, 48 patients (15% of 322 patients; please see also Table 2) were self-identified as having RLS, 33% could be categorized as having (possible) RLS, and 215 (67%) reported no RLS. Yet, when asked whether the presence or absence of RLS was evaluated by a physician, only 15% of the patients responded in the affirmative. Of these patients, 45% received a treatment for their RLS (N = 21/47, 95% CI: 31–59%). An analysis of physician-diagnosed RLS and the RLS-DI, revealed a relative frequency of RLS in HHT patients of 22% to 36% (Table 2, calculations a and b). This value is markedly higher than the historical RLS percentage of 0.8% to 18% in the general population. In fact, 8% (N = 25/322, 95% CI: 0.05–0.11) of the HHT patients had RLS which was not already been diagnosed by a physician. In addition, when “possible RLS” is included, this number increases to 21% (N = 68/322, 95% CI: 17–26%). Therefore, according to our data, between 8% and 21% of the patients with HHT might be underdiagnosed with RLS. These 25 and 68 patients equal 35% and 59%, respectively of the RLS patients of this population (of 72 and 115 patients respectively, when including “possible RLS”; please see also Table 2, calculations a and b). In 8 cases, the patients who had been diagnosed with RLS by their physician before, did not reach a high score in the RLS-DI, possibly due to efficient therapy by medication.

For analyzing the influence of various HHT symptoms and the need of different treatment options, we analyzed patients who we assumed, according to their physician or the RLS-DI, might suffer from RLS (“assumed RLS”; see Materials and Methods section). About one fifth of the HHT patients with anemia also suffered from RLS (N = 63/299, 21%), and patients who had gastrointestinal bleeding (GI) showed an increased prevalence of RLS. In line with this, the patients who needed iron supplementations were more often diagnosed with a neurological disorder. Regarding the diagnosis of RLS, men and women were equally affected and there was no difference in patients’ region of origin or genetics (Table 3).

## 4. Discussion

The historical literature supports an association between HHT and an increased prevalence of neurological disorders. In particular, the reduced pulmonary function as a blood filter in patients with pulmonary arteriovenous malformations, may lead to migraines, cerebral abscesses and strokes [8,17]. Patients with HHT often suffer from recurrent bleeding resulting in a CIDA, which is a recognized cause for restless legs syndrome. However, to our knowledge a potential association between HHT and RLS has not been studied. 

In the adult general population, the prevalence of RLS is roughly 0.8–18% [18,19,20]. Taking into account the patients who had already been diagnosed with RLS and those with a pathologic test result in the RLS−DI, the calculated prevalence of 22–36% for HHT was higher than in the general adult population. According to the literature, about one fourth to one fifth of patients with CIDA exhibit clinically significant RLS [21]. Knowing that CIDA is a common comorbidity in patients with HHT [8] indicates that to diagnose this disease, it might be necessary to ask specific questions about restless legs. Ferri et al. postulated a single question for the rapid screening of RLS (RLS screening: “When you try to relax in the evening or sleep at night, do you ever have unpleasant, restless feelings in your legs that can be relieved by walking or movement?”, 100% sensitivity and 96.8% specificity) [22]. In addition, patients with RLS and CIDA often suffer from a sleep-related morbidity [21]. They reported decreased sleep times and had a higher risk for complications like cardiovascular disease [23] and immunological impairment [24]. In our study half of the patients perceived a sleep disorder, but only 5% got further testing. Thus, RLS in HHT is likely underdiagnosed and undertreated. A screening for RLS in HHT patients, especially with CIDA, might be important to prevent comorbidities.

The most common conditions associated with RLS include iron deficiency with low serum ferritin levels, and subsequent low central nervous system intracellular iron [25]. Previous studies indicated that iron deficiency reduces cerebral dopamine receptors and transporters [26]. Alterations in the dopaminergic system may lead to RLS [27]. Anemia is defined by low hemoglobin levels, and patients’ iron stores can be measured best by using serum ferritin, indicating the need for iron supplementation [28,29]. In the event of infectious diseases, normal or high ferritin levels do not exclude iron deficiency, as it also acts as an acute phase protein. Here, serum soluble transferrin receptor provides a valuable addition to existing methods, although it is known to be a less specific indicator for iron deficiency than ferritin [30]. Typically, patients with iron deficiency (anemia) aim for normal iron stores and hemoglobin levels. For RLS with CIDA, the iron status is likely to be more important than the anemia [21]. Accordingly, iron treatment reduced RLS symptoms in patients with low serum ferritin levels [26]. Therefore, to treat CIDA in patients with HHT, hemoglobin levels, serum ferritin and maybe soluble transferrin receptors, are important diagnostic markers and should be routinely measured [31]. However, in accordance with another study about CIDA in patients with RLS [21], regarding the RLS diagnosis in patients with HHT there was no relationship between the degree of anemia, iron supplementation, or the need for blood transfusions.

In our study, males and females were equally affected by anemia and RLS. Both, HHT and RLS have been associated with a reduced quality of life and an increased mortality rate [25,32,33]. Treating the underlying condition in CIDA RLS with iron supplementation improves patients’ restless legs [34]. Increasing awareness to, or even screening for RLS due to iron deficiency, may therefore improve quality of life and life expectancy of patients with HHT.

Methodological limitations of online-based survey studies warrant consideration. However, multicenter studies with sufficient numbers of patients with HHT are sparse. By questionnaire, patient data obtained from diverse countries can be collected. However, the RLS-DI was not validated for this setting. As we used an English and German version of the survey, most respondents came from North America and Western Europe. Language barriers may be the reason why patients with HHT may not have responded to the survey. Not all patients answered all questions, resulting in a relatively low response rate and smaller subgroups for the analysis. Being an autosomal dominant inherited disease, men and women are equally affected by HHT. In this study, more female patients answered the survey. In general, women are also more prone to RLS than men [35]. In addition, regional differences in the prevalence of RLS exist. Most patients who answered the questionnaire came from Europe and North America where an increased frequency of RLS was reported [36]. We asked if the patients had already been diagnosed with RLS and included the RLS-DI. Other factors also causing RLS (e.g., medication, rheumatoid arthritis, pregnancy, neuropathy and fibromyalgia), iron parameters (e.g., ferritin- or transferrin receptor-level), or a classification of the severity of RLS, were not recorded, and the answers of patients, or diagnoses made by their physicians might have been inaccurate. The RLS-DI is a tool to diagnose actual and persistently present RLS in patients of a sleep laboratory population [14]. The purpose of this study was not to analyze data of sleep laboratories. As we aimed to reach a high number of patients with HHT from all over the world, we relied on patient self-reporting. No direct patient interviews, physical examinations, or evaluation of medical records was performed.

## 5. Conclusions

In conclusion, we showed for the first time that RLS in patients with HHT is an underdiagnosed and undertreated condition. In particular, patients with recurrent epistaxis and gastrointestinal bleeding had a higher prevalence of RLS. Regular assessment of CIDA parameters and questions about restless legs, as presented in the RLS-DI or mentioned above in patients with HHT, could increase the diagnostic detection probability. Further data are necessary to evaluate if, and what type of screening for RLS in HHT may be justified. Iron supplementation may resolve symptoms of RLS. Thus, paying attention to RLS especially in patients with CIDA may improve patients’ quality of life, and even their mortality rate.

## Figures and Tables

**Figure 1 jcm-10-01993-f001:**
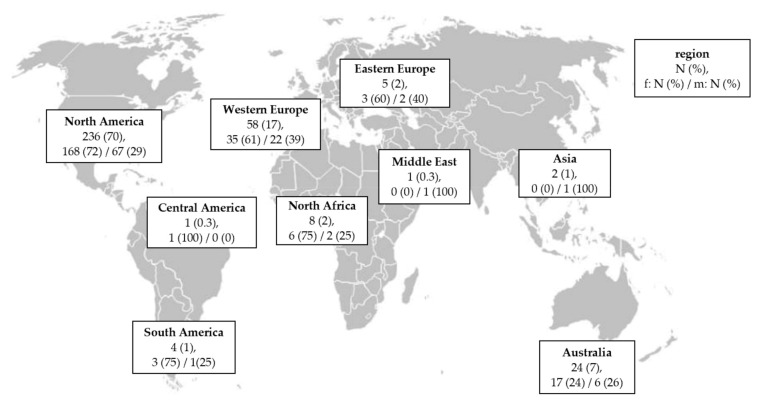
Geographic data of patients with HHT.

**Figure 2 jcm-10-01993-f002:**
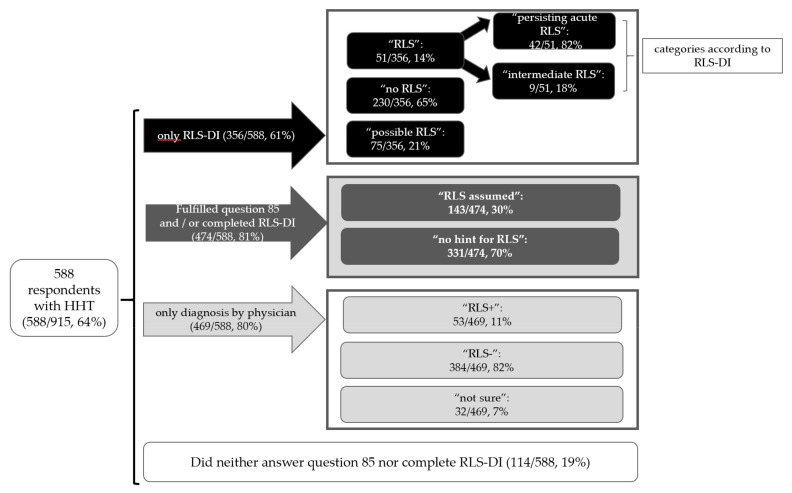
Diagnostic assignment of RLS.

**Table 1 jcm-10-01993-t001:** Sex and genetic results of patients with HHT.

	Number of Answered Questions (*n* (%))		Number of Patients (*n* (%))
**sex**	467/474 (99)	males	148 (32)
		female	319 (68)
**genetic testing**	468/474 (99)	yes	260 (56)
		no	208 (44)
**genetic** **mutation**	190/474 (40)	HHT Type 1	72 (38)
		HHT Type 2	114 (60)
		SMAD 4	3 (2)
		HHT Type 5	1 (1)

**Table 2 jcm-10-01993-t002:** RLS-DI questionnaire versus physician diagnosis of RLS.

		Diagnosis by RLS-DI			
		Yes	Possible	No	Sum
**RLS diagnosis by** **physician**	**yes**	23	16	8	47 ^a,b^
	**no**	25 ^a,b^	43 ^b^	207	275
	**sum**	48	59	215	322

There were 322 patients with hereditary hemorrhagic telangiectasia (HHT) who answered both, the RLS-DI and the question about the RLS diagnosis by their physician (322/474, 68%). Only the RLS-DI: RLS (“yes”): 48 patients (15%, 95% CI: 12–19%), “possible” RLS: 59 patients (18%, 95% CI: 14–23%), and “no” RLS: 215 patients (67%, 95% CI: 61–72%. Only RLS diagnosis by physician (Appendix A): RLS+ (“yes”): 47 patients (15%, 95% CI: 12–19%), or (“no”): 275 patients (85%, 95% CI: 81–82%). ^a^ Number of HHT patients with RLS tabulated in the calculation above: (47 + 25)/322 = 72/322 = 22%, 95% CI: 18–27%; of these 72, only 47 (65%, 95% CI: 54–75%) had already been diagnosed by a physician, and 35% (25/72, 95% CI: 25–46%; 8% of 322, 95% CI: 5–11%) had not, respectively. ^b^ Number of HHT patients with definite and possible RLS counted in the calculations above: (47 + 25 + 43)/322 = 115/332 = 36%, 95% CI: 0.31–0.41; of these 115, only 47 (41%, 95% CI: 32–50%) had already been diagnosed by a physician, and 59% (68/115, 95% CI: 50–68%; 21% of 322, 95% CI: 17–26%) had not, respectively.

**Table 3 jcm-10-01993-t003:** Influence of HHT symptoms and patients’ characteristics on the prevalence of self-reported RLS.

Symptom	Patients	Assumed RLS *	Percent	PR	95%CI
**gastrointestinal bleeding**					
no	115	13	11	Ref.	
yes	144	44	31	2.70	1.53–4.77
**epistaxis**					
no	13	1	8	Ref.	
yes	399	80	20	2.61	0.39–17.31
**anemia**					
no	71	15	21	Ref.	
yes	299	63	21	1.00	0.60–1.64
**blood transfusions**					
no	180	31	17	Ref.	
yes	125	41	33	1.90	1.27–2.86
**iron intake** **(oral and/or intravenous)**					
no	67	7	10	Ref.	
yes	345	74	21	2.05	0.99–4.26
**genetic mutation ^†^**					
type 1	59	17	29	1.34	0.77–2.31
type 2	102	22	22	Ref.	
**Origin ^†^**					
America	213	56	26	Ref.	
Africa	8	4	50	1.90	0.92–3.94
Australia	20	4	20	0.76	0.31–1.88
Europe	52	8	15	0.59	0.30–1.15
**sex**					
women	274	55	20	Ref.	
men	132	23	17	0.87	0.56–1.35

A total of 474 patients with hereditary hemorrhagic telangiectasia (HHT) answered the RLS−DI and/or if they were diagnosed as having RLS by their physicians. * HHT patients with “assumed RLS”, patients who had been diagnosed with RLS, and those as yet undiagnosed with RLS who had pathological results in the RLS−DI (at least “possible RLS” according to the RLS−DI; N = 143/474, 30%; please also see Figure 2). ^†^ Because of the small numbers, patients with SMAD4 mutations (N = 1), HHT type 5 (N = 1), and those from Asia (N = 3), were excluded. A log-binominal regression analysis was performed. The number of all patients (Patients) and patients with an assumed diagnosis of RLS (RLS+) are shown. RLS = Restless Legs Syndrome, DI = diagnostic inventory, PR = prevalence ratio, 95% CI = 95% confidence interval, Ref = Reference.

## Data Availability

Anonymized data will be shared by request from any qualified investigator.

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
