# Peer review of "Restless Leg Syndrome Is Underdiagnosed in Hereditary Hemorrhagic Telangiectasia—Results of an Online Survey"

_jcm, 2021, doi:10.3390/jcm10091993_

Round 1

Reviewer 1 Report

Comments to the Authors.

In the manuscript entitled “Restless Leg Syndrome is underdiagnosed in Hereditary Hemorrhagic Telangiectasia”, Droege and co-authors performed an online survey additioning 82 questions related to HHT to the standardized RLS-Diagnostic Index questionnaire, with the aim to evauate the association between HHT-related symptoms and the prevalence of RLS.
The proposed topic is innovative and interesting, as the survey developed by the authors could represent a useful complementary tool for the diagnosis of RLS in patients with HHT, as weel as to prevent chronic iron deficiency anemia in these patients.
Although the work is well designed, the statistical analysis satisfactory and the list of references pertinent and exhaustive, some minor revisions need to be addressed before to have the manuscript accepted for publications:
1) Line 86 - Please add at the first line of the Results section: “Diagnosis of HHT was…”
2) Line 93 – Where does the number 467 come from? In total, HHT patients doesn’t were 474? In the previous lanes has been reported that “the data refers to 474 patients with HHT”. Could the authors explain better and clarify, please.
3) Lanes 104-105 – The sentence “Most patients came from North America and western Europe (N = 294/339, 87%)”, is the same already reported in lines 94-95. Probably, lines 104-105 are comprised in the figure legends. Is this correct?
4) Line 111 – Delete “patients” after HHT.
5) Table 2: in table caption add the GIB acronym and its full name.
6) The Results section, as it is structured, is quite “heavy” to deal with; a lot of not always relevant data are reported, expressed as percentage and numbers, not always clear enough, making the reader’s understanding rather intricate. If possible, try to make this chapter most fluent for a more enjoyable reading.

Author Response

Dear Professor Andrès, dear Doctor Cuesta, dear Editors, dear Reviewer and dear Sir or Ma’am,

We would like to thank the reviewer for the comprehensive review of our manuscript and have inserted our point-by-point response as requested: 

In the manuscript entitled “Restless Leg Syndrome is underdiagnosed in Hereditary Hemorrhagic Telangiectasia”, Droege and co-authors performed an online survey additioning 82 questions related to HHT to the standardized RLS-Diagnostic Index questionnaire, with the aim to evauate the association between HHT-related symptoms and the prevalence of RLS.

The proposed topic is innovative and interesting, as the survey developed by the authors could represent a useful complementary tool for the diagnosis of RLS in patients with HHT, as well as to prevent chronic iron deficiency anemia in these patients.

àWe thank the reviewer for this comment.

Although the work is well designed, the statistical analysis satisfactory and the list of references pertinent and exhaustive, some minor revisions need to be addressed before to have the manuscript accepted for publications:

1) Line 86 - Please add at the first line of the Results section: “Diagnosis of HHT was…”

à As recommended we added „of HHT“ (line 65).

2) Line 93 – Where does the number 467 come from? In total, HHT patients doesn’t were 474? In the previous lanes has been reported that “the data refers to 474 patients with HHT”. Could the authors explain better and clarify, please.

à 467 patients answered the question about their sex. Unfortunately, not all patients answered every question of the survey probably due to its length. Therefore, with reporting the number of patients who answered each question we addressed this problem (please see also lines 98-99 (results) and 282 ff. (discussion), marked version).

3) Lines 104-105 – The sentence “Most patients came from North America and western Europe (N = 294/339, 87%)”, is the same already reported in lines 94-95. Probably, lines 104-105 are comprised in the figure legends. Is this correct?

à Yes, these lines belong to the figure legend. We changed the layout and hope that it’s easier to read now. 

4) Line 111 – Delete “patients” after HHT.

à The reviewer is right, we deleted the word „patients“.

5) Table 2: in table caption add the GIB acronym and its full name.

à We thank the reviewer for this comment and added the full name in table 2.

6) The Results section, as it is structured, is quite “heavy” to deal with; a lot of not always relevant data are reported, expressed as percentage and numbers, not always clear enough, making the reader’s understanding rather intricate. If possible, try to make this chapter most fluent for a more enjoyable reading

à The reviewer is correct. Therefore, we added subheadings and restructured the results section as also recommended by reviewer 2. In addition, Figure 1 with data about stratification of diagnostic assignments was implemented. We hope now this section is more fluent for a more enjoyable reading.

We hope we have addressed all the questions and concerns of the reviewers, and we thank you and the reviewer for the time and feedback. Using the “track changes” function the text in the revised manuscript that has been added or modified is highlighted. Please let me know if there is any other information or clarification I can provide at this time.

Sincerely

Dr. Freya Droege, and on the behalf of the co-authors

Reviewer 2 Report

General comments

The article is a bit confusing with various numbers of patients, and a lack of precision in the methods section.

Since it is the core subject, I would have appreciated the methods section to be more specific about RLS. Maybe the authors should talk about how was the question on sleep disorders/diagnosis formulated, explain that there was a question on RLS diagnosis by a physician; And somewhere explain that the objective was to know what proportion of patients presenting RLS symptoms (according to RLS-DI) were actually diagnosed by a physician. Moreover, a confidence interval for this percentage seems mandatory.

Results section should start with the total number of “invitation” and the total number of respondents, and the total number having completed the questionnaire, so that an overall response rate would be available. Please provide a flow chart that would allow to navigate between the different subpopulation mentioned in the article.

The authors should divide the results section into themes to allow better comprehension (population / HHT symptoms / sleep disorders&RLS / exlicative analysis)

Specific comments

  • Add the study type in title : survey
  • Follow the CHERRY guideline for surveys
  • Precise the estimated time for fulfilling the questionnaire and drop-out rate
  • Line 86 : did the authors asks for all items of the curacao criteria in the questionnaire ? moreover this information should be in the methods section, not in the results
  • How many emails were sent?
  • Is 915 corresponding to all participants having started the questionnaire? Having finished the questionnaire?
  • How is it possible that 40% of patients did not have HHT whereas the questionnaire was sent to HHT networks?
  • Line 93 to 100: not easy to read, maybe put in a table
  • The world map is not very useful, since the origin of patients is more related to the networks involved in the study. The figures are not explained, what does 236 (70, 168(72), 67 (29) means? If conserved, it has to be more precise
  • Line 103: I don’t understand where does 339 come from, as well as other figures in the paragraph
  • Line 109 : please precise in the methods what kind of question allowed to say that patients “reported that they suffered from a sleep disorder”;
  • Line 110 : please precise in the methods what kind of question allowed to say that patients “have been screened for sleep-related disorder”;
  • Line 111: the authors mention that 322 patients with HHT have completed the section on RLS whereas on line 92 they said it was 474 patients
  • Table 1 : 332 is mentioned, I guess it is a typo
  • Please provide confidence interval; and add in the text the number and percentage of underdiagnosed patients since it is the title of the article
  • Some interesting results on the core theme are provided and explained in the abstract but not in the results section- please add
  • For table 1 and figure 1: it is difficult to distinguish text from legend
  • Starting line 137: the whole paragraph will be better suited at the beginning of the results section
  • Line 138: here the authors mentioned 473 instead of 474. Typo? If not, the authors should think about defining homogeneous subpopulation for each theme even if this leads to deleting some incomplete patients.
  • Line 149: here the authors mentioned patients suffering from RLS (299)—according to which definition. this does not seem to match with table 1
  • Line 149-150 please provide pvalue to support the results. it does not seems that recurrent epistaxis is significantly associated with RLS
  • Did the authors perform a multivariate analysis since all the studied parameters are correlated?
  • Table 2 : either write GIT in full, or use the same abbreviation as in the text (GI)
  • Table 2 : rephrase the first sentence of legend. I don’t understand; and this does not match with the explanation in line 91-92
  • Definition of RLS+ should be provided in the methods section
  • The text of table 2 legends would be useful in the text and maybe more interestingly provided to supplement table1
  • Table 2 mention genetic mutation type 1 and 2 whereas the introduction mentions ENG and ALK1
  • Line 180 : the authors propose a simple question to diagnose RLS, was this question available in the questionnaire?
  • Line 236-237: it does not seems that recurrent epistaxis is significantly associated with RLS
  • Abstract :again the parameters mentioned do not seem to be significantly associated
  • Not sure that the 42 references are necessary, maybe limit to 30
  • Add the questionnaire as supplemental information

Author Response

Dear Professor Andrès, dear Doctor Cuesta, dear Editors, dear Reviewer and dear Sir or Ma’am,

We would like to thank you for your swift and comprehensive response to our manuscript on Restless Leg Syndrome in Hereditary Hemorrhagic Telangiectasia. The comments were very helpful in improving our manuscript. We are happy that your journal supports research on rare diseases and is interested in the topic of our study. We revised this manuscript in response to your and the reviewer’s comments:

The article is a bit confusing with various numbers of patients, and a lack of precision in the methods section.

Since it is the core subject, I would have appreciated the methods section to be more specific about RLS. Maybe the authors should talk about how was the question on sleep disorders/diagnosis formulated, explain that there was a question on RLS diagnosis by a physician; And somewhere explain that the objective was to know what proportion of patients presenting RLS symptoms (according to RLS-DI) were actually diagnosed by a physician. Moreover, a confidence interval for this percentage seems mandatory.

  • We thank the reviewer for this comment and improved our methods section according to it. By adding the supplementary data, all questions analyzed in this study could be published. As suggested, confidence intervals for percentages were also implemented.

Results section should start with the total number of “invitation” and the total number of respondents, and the total number having completed the questionnaire, so that an overall response rate would be available. Please provide a flow chart that would allow to navigate between the different subpopulation mentioned in the article.

  • In the beginning of the results section (now lines 65-66, marked version), we cited other manuscripts in which the stratification diagnostic assignments were pictured. But the reviewer is correct, the various numbers of patients and different RLS diagnostic tools stated in the results section might be a bit confusing. Therefore, we added Figure 1 with the different subpopulations mentioned in the article.

The authors should divide the results section into themes to allow better comprehension (population / HHT symptoms / sleep disorders&RLS / exlicative analysis)

  • We thank the reviewer for this comment and added subheadings as recommended.

Specific comments

Add the study type in title : survey

Follow the CHERRY guideline for surveys

Precise the estimated time for fulfilling the questionnaire and drop-out rate

  • We added the study type in the title and added Figure 1 so that the drop-out rate is recognizable. According to the CHERRY guidelines we discussed this study limitation in lines 282-283 (marked version).

Line 86: did the authors asks for all items of the curacao criteria in the questionnaire ? moreover this information should be in the methods section, not in the results

  • Yes, as now stated in the materials and methods section, the diagnosis of HHT was established using the modified Curaçao criteria as published by Hosman et al.[13] and Droege et al.[14]. In the supplementary data the survey can be found.

How many emails were sent?

  • As now reported in lines 99-102 (marked version), 334 of the 588 patients (57%) could be contacted via mail and 105 of those answered.

Is 915 corresponding to all participants having started the questionnaire? Having finished the questionnaire?

How is it possible that 40% of patients did not have HHT whereas the questionnaire was sent to HHT networks

  • The reviewer is correct, there are quite a lot missing values probably due to the length of the questionnaire. We also discussed this in the study limitations at the end of the article. In 334 cases the patients gave us their mail address. Thus, we could ask for more information. However, in most cases the questions were not answered by all patients. With reporting the number of patients who answered each question we addressed this problem.

Line 93 to 100: not easy to read, maybe put in a table

The world map is not very useful, since the origin of patients is more related to the networks involved in the study. The figures are not explained, what does 236 (70, 168(72), 67 (29) means? If conserved, it has to be more precise

  • We are thankful for this comment and added data about patients’ sex and genetic results in Figure 2. Thus, the text should be easier to read.

We believe that the world map illustrates quite well that with the help of different patient organizations (see acknowledgements) data from patients from all over the world were analyzed. But we agree with the reviewer that the Figure 2b is not well explained. Therefore, we improved the explanation in the legend. However, if also the editors prefer to replace the map by a short table we would of course be willing to do so.

Line 103: I don’t understand where does 339 come from, as well as other figures in the paragraph

  • As mentioned above, unfortunately, not all patients answered every question of the survey probably due to its length and the need of medical reports. If we would analyze only the data from patients who completed every question of the survey very few data would have been left over. Therefore, with reporting the number of patients who answered each question we addressed this problem.

Line 109 : please precise in the methods what kind of question allowed to say that patients “reported that they suffered from a sleep disorder”;

Line 110 : please precise in the methods what kind of question allowed to say that patients “have been screened for sleep-related disorder”;

  • As recommended, the survey can be found in the supplementary data. In question 95 (“Do you suffer from a sleeping disorder (like Sleep Apnea/falling asleep or continual sleepiness)?”) patients answered if they suffered from any sleep disorder. In question 98 they reported if they had been screened for it (“Have you ever been screened for sleeping disorders, (for example, apolysomnography = test while sleeping), and if so, did this test reveal any signs of Restless Legs Syndrome? (see clinic papers)”). We added the number of the questions in the text so that they can be easily found in the supplementary data.

Line 111: the authors mention that 322 patients with HHT have completed the section on RLS whereas on line 92 they said it was 474 patients

  • 474 patients with HHT completed responses for questions pertaining to RLS and 322 patients answered the RLS-DI. We are sorry for the confusion and rewrote lines 166-167 (marked version, please see also Figure 1).

Table 1: 332 is mentioned, I guess it is a typo

  • We thank the reviewer for this comment and corrected this number.

Please provide confidence interval; and add in the text the number and percentage of underdiagnosed patients since it is the title of the article

  • As suggested, we added the confidence intervals of percentages. Moreover, the percentage of underdiagnosed patients was emphasized in the results section. These data can be found in lines 182-183 (marked version). As this is the core theme of our article, we now emphasized these findings and thank the reviewer for this comment.

Some interesting results on the core theme are provided and explained in the abstract but not in the results section- please add

  • We think all results are provided in the both sections and we improved the explanations in the results sections. For a better overview for the reviewer all results listed in the abstract are marked in blue in the results section.

For table 1 and figure 1: it is difficult to distinguish text from legend

  • The reviewer is correct. We hope text and legend can be distinguished now

Starting line 137: the whole paragraph will be better suited at the beginning of the results section

  • The reviewer is right, we moved this part further up (please see lines 146 ff, marked version).

Line 138: here the authors mentioned 473 instead of 474. Typo? If not, the authors should think about defining homogeneous subpopulation for each theme even if this leads to deleting some incomplete patients.

  • No, it is not a typo. 473 patients answered the question about their disease progression. Unfortunately, not all patients answered every question of the survey probably due to its length. Therefore, with reporting the number of patients who answered each question we addressed this problem (please see also lines 98-99 (results, marked version) and 282-283 (discussion, marked version)).

Line 149: here the authors mentioned patients suffering from RLS (299)—according to which definition. this does not seem to match with table 1

  • The reviewer is correct, we needed to improve the explanation of the chosen definitions. We did so by adding Figure 1 and an extra explanation in the methods section. Please see also lines 209 ff. (marked version). Again, we thank the reviewer for this comment.

Line 149-150 please provide pvalue to support the results. it does not seems that recurrent epistaxis is significantly associated with RLS

  • We quantified the strength of association between bleeding parameters (epistaxis, GI bleeding, ….) and RLS by use of log-binomial models that provide prevalence ratios and corresponding 95% confidence intervals (see Table 2). We avoided the dichotomization into significant and non-significant statistical associations for several reasons (Wasserstein RL, Lazar NA. The ASA's Statement on p-Values: Context, Process, and Purpose. Am Statistician 2016; 70:2, 129-133).

According to Lash, epidemiologic research is an exercise in measurement. Its objective is to obtain a valid and precise estimate of either the occurrence of disease in a population or the effect of an exposure on the occurrence of disease.” (Lash TL: Heuristic thinking and inference from observational epidemiology. Epidemiology 2007;18:67-72).

We are calculating and reporting confidence intervals to assess the precision of our estimates because our goal is estimation and not significance testing. We wish to avoid publication bias by preferential reporting of significant results. Instead, we judge the value of our estimates by their precision and validity.

However, the reviewer is able to re-translate confidence intervals into significant and non-significant by checking whether the point estimate of 1.0 is included (=non-significant) or excluded (significant) from the confidence interval.

Did the authors perform a multivariate analysis since all the studied parameters are correlated?

  • Not all the participants answered all the questions and this resulted in various subgroups with many missing values. Thus, the authors decided not to perform a multivariate analysis since its interpretation will be limited.

Table 2 : either write GIT in full, or use the same abbreviation as in the text (GI)

  • As recommended we added „gastrointestinal bleeding“ in table 2 and deleted “GIT”.

Table 2 : rephrase the first sentence of legend. I don’t understand; and this does not match with the explanation in line 91-92

  • We believe that thanks to the remarks of the reviewer we could optimize the explanation. We improved the legend and if you take additionally the explanation in the methods section and Figure 1 into account, the definitions and numbers should be clear. If not, please let us know on which point we need to further improve our statement.

Definition of RLS+ should be provided in the methods section

  • The reviewer is right, we added this information in the methods section.

The text of table 2 legends would be useful in the text and maybe more interestingly provided to supplement table1

  • The reviewer is correct, we relocated parts of the text of the legend in the text and improved the legend.

Table 2 mention genetic mutation type 1 and 2 whereas the introduction mentions ENG and ALK1

  • We are thankful for this remark and added the mutation types in the introduction section.

Line 180: the authors propose a simple question to diagnose RLS, was this question available in the questionnaire?

  • No, unfortunately this question was not available in our questionnaire. Further studies should investigate if this simple screening-question could be also used in patients with HHT.

Line 236-237: it does not seems that recurrent epistaxis is significantly associated with RLS

Abstract :again the parameters mentioned do not seem to be significantly associated

  • Among the bleeding parameters, epistaxis was the parameter that produced the most imprecise estimate of the prevalence ratio (PR=2.61, 95%CI 0.39 – 17.31). We therefore revised the abstract and now present only estimates for GI bleeding, iron intake and blood transfusions that are considerably more precise. We also revised the text in the result section accordingly.

Not sure that the 42 references are necessary, maybe limit to 30

  • As the reviewer suggests we reduced the number of references.

Add the questionnaire as supplemental information

  • As recommended, we added it as supplementary data.

We hope we have addressed all the questions and concerns of the reviewers, and we thank you and the reviewer for the time and feedback. Using the “track changes” function the text in the revised manuscript that has been added or modified is highlighted. Please let me know if there is any other information or clarification I can provide at this time.

Sincerely

Dr. Freya Droege, and on the behalf of the co-authors

Round 2

Reviewer 2 Report

The manuscript has improved thanks to definition provided in the methods section, but there are still some parts that are difficult to understand

Line 73 +line 129-130: The methods specify that the diagnosis is made according the curacao criteria but there is no mention in the questionnaire. The only questions is “Have you been diagnosed with HHT”, I cannot see how the authors could ascertain that diagnosis was based on curacao

Line 102 should start by explaining the figure 915.

Line 121: the authors mentioned here that some patients were contacted to complete missing values. I didn’t catch this information on the first version. Maye the authors should add these information in the methods, I am not sure that the figures are useful. Moreover, they refer to figure 1 but there is no mention of theses “completed missing data”, the authors should keep it simple. How many emails were sent: the authors misunderstood my question which was: how many patients were contacted in order to reach 915 answers?

Line 125 : not sure why the figure appears here while it is related with line 198 (“Sleep related disorders in patients with HHT”)

Page 4: add the figure 915 at the beginning of the flow chart

Figure 1 is a mix of a study flow chart and results on sleep disorder. Maybe the authors should think about a flow chart containing the following information: 915 / 312-588-15 (diagnosis of HHT) /474-114 (evaluation of sleep available).

Figure 2: information on sex and genetic testing would be more appropriate in a table

Harmonize units between estimates and CI (10% or 0.10)

“Using the RLS-DI, 48 patients (15% of 322 patients, 95% CI: 0.11 – 0.19; of those intermittent RLS: N = 9/48 equalling 19%, 95% CI: 0.10 – 0.32and persistent RLS: N = 39/48 equalling 81%, 95% CI: 0.68 –0.90) were self-identified as having RLS, 33%could be categorized as having (possible) RLS (“RLS” as reported above and “possible RLS”: N = 107/322, 95% CI: 0.28 – 0.39; possible RLS: N = 59/322,18%, 95% CI: 0.14 – 0.23) and 215 (67%, 95% CI: 0.61 – 0.72) reported no RLS. Yet, when asked whether the presence or absence of RLS was evaluated by a physician, only 15% (47/322, 95% CI: 0.11 – 0.19) of the patients responded in the affirmative. 45% of these patients received a treatment for their RLS (N = 21/47, 95% CI: 0.31 –0.59).” This paragraph is barely understandable, the authors should simplify the writing, delete some CI (as they are not necessarly useful for each percentage), provide links with figure 1 (if pertinent)

Author Response

2nd Revision Letter (JCM-1176811):

Restless Leg Syndrome is underdiagnosed in Hereditary Hemorrhagic Telangiectasia

Dear Professor Andrès, dear Doctor Cuesta, dear Editors, dear Reviewer and dear Sir or Madam,

We thank the reviewer for her/his comments and revised our manuscript:

The manuscript has improved thanks to definition provided in the methods section, but there are still some parts that are difficult to understand

Line 73 +line 129-130: The methods specify that the diagnosis is made according the curacao criteria but there is no mention in the questionnaire. The only questions is “Have you been diagnosed with HHT”, I cannot see how the authors could ascertain that diagnosis was based on curacao

  • There are several questions in the supplementary data that represent the Curaçao Criteria, however, they are not directly next to each other: Questions number 4 (family history), number 21 (telangiectasia), numbers 24-26 (organ malformations) and number 28 (epistaxis).

Line 102 should start by explaining the figure 915.

  • Thank you for this hint: 915 persons responded to the survey after different patient organizations had informed their members. We added this to the results section (lines 98-100).

Line 121: the authors mentioned here that some patients were contacted to complete missing values. I didn’t catch this information on the first version. Maye the authors should add these information in the methods, I am not sure that the figures are useful. Moreover, they refer to figure 1 but there is no mention of theses “completed missing data”, the authors should keep it simple. How many emails were sent: the authors misunderstood my question which was: how many patients were contacted in order to reach 915 answers?

  • We thank the reviewer for this comment and added information on emails sent by the authors to patients in the methods section. As mentioned above, we contacted different patient advocacy groups, which informed their members about the study. Therefore, initially, the authors did not contact HHT patients directly. At the end of the survey, respondents could provide their email address so that the authors could contact them in case of back queries.

Line 125 : not sure why the figure appears here while it is related with line 198 (“Sleep related disorders in patients with HHT”)

  • Figure 2 in line 125 presents general data of the study population, which are also described in lines 121ff. The part about “sleep related disorders in patients with HHT” was also represented in Figure 1 and therefore the reviewer is correct. Thus, we modified former Figure 1 and inserted it further down as figure 2. In order to describe the stratification of diagnostic assignments we referred to an already published figure instead (see also methods section line 67, Droege F. et al.. Nasal self-packing for epistaxis in Hereditary Hemorrhagic Telangiectasia increases quality of life. Rhinology 2019, 57, 231-239, doi:10.4193/Rhin18.141). The now modified Figure can be found as Figure 2. We hope that the deleted former figure 1 – which is now figure 2 – does not show up in its former position in the text due to conflicting track changes features of different Microsoft Word versions. If this should be the case, we will be happy to also provide a pdf version for clarification.

Figure 1 is a mix of a study flow chart and results on sleep disorder. Maybe the authors should think about a flow chart containing the following information: 915 / 312-588-15 (diagnosis of HHT) /474-114 (evaluation of sleep available).

  • The reviewer is right. As mentioned above we modified this figure (please see comment above). Therefore, now the numbers 915/ 588/ 474-114 are mentioned. For additional information about the numbers 915/ 312- 588- 15 we included the reference to a figure of another paper which exactly explains these numbers. Unfortunately, we can only upload text in the answer letter. However, for your interest we also uploaded the figure and the complete article under the following link: https://cloud.uk-essen.de/d/8f9190f3082e4a3bb368/

Password: 12345678

Page 4: add the figure 915 at the beginning of the flow chart.

  • Please see comments above.

Figure 2: information on sex and genetic testing would be more appropriate in a table

  • As recommended, we put the information on sex and genetic testing in a table.

Harmonize units between estimates and CI (10% or 0.10)

  • We agree with the reviewer and harmonized the data accordingly.

“Using the RLS-DI, 48 patients (15% of 322 patients, 95% CI: 0.11 – 0.19; of those intermittent RLS: N = 9/48 equalling 19%, 95% CI: 0.10 – 0.32and persistent RLS: N = 39/48 equalling 81%, 95% CI: 0.68 –0.90) were self-identified as having RLS, 33%could be categorized as having (possible) RLS (“RLS” as reported above and “possible RLS”: N = 107/322, 95% CI: 0.28 – 0.39; possible RLS: N = 59/322,18%, 95% CI: 0.14 – 0.23) and 215 (67%, 95% CI: 0.61 – 0.72) reported no RLS. Yet, when asked whether the presence or absence of RLS was evaluated by a physician, only 15% (47/322, 95% CI: 0.11 – 0.19) of the patients responded in the affirmative. 45% of these patients received a treatment for their RLS (N = 21/47, 95% CI: 0.31 –0.59).” This paragraph is barely understandable, the authors should simplify the writing, delete some CI (as they are not necessarly useful for each percentage), provide links with figure 1 (if pertinent)

  • The reviewer is correct, the paragraph is misunderstandable therefore we changed it. However, the respective link refers to table 2.

We hope that the changes meet your intentions and further clarify the manuscript. As requested, modified or added parts of the text were marked either by using the “track changes” function and/or by marking it yellow in the revised manuscript. Please do not hesitate to let us know if further information or clarification should be desirable – we would be happy to provide these.

Sincerely

Dr. Freya Droege, and on the behalf of the co-authors
